# Direct Observations of the Structural Properties of Semiconducting Polymer: Fullerene Blends under Tensile Stretching

**DOI:** 10.3390/ma13143092

**Published:** 2020-07-10

**Authors:** Mouaad Yassine Aliouat, Dmitriy Ksenzov, Stephanie Escoubas, Jörg Ackermann, Dominique Thiaudière, Cristian Mocuta, Mohamed Cherif Benoudia, David Duche, Olivier Thomas, Souren Grigorian

**Affiliations:** 1Aix Marseille Univ, U. Toulon, CNRS, IM2NP (Institut Matériaux Microélectronique et Nanosciences de Provence), Campus St-Jérôme, 13397 Marseille CEDEX 20, France; mouaad-yassine.aliouat@im2np.fr (M.Y.A.); david.duche@univ-amu.fr (D.D.); olivier.thomas@im2np.fr (O.T.); 2Ecole Nationale Supérieure des Mines et de la Métallurgie, L3M, Annaba, Sidi Amar W129, Algeria; mohamed-cherif.benoudia@ensmm-annaba.dz; 3Institute of Physics, University of Siegen, D-57068 Siegen, Germany; ksenzov@physik.uni-siegen.de; 4Aix-Marseille University, CNRS, CINAM, 13007 Marseille, France; jorg.ackermann@univ-amu.fr; 5Synchrotron SOLEIL, L’Orme des Merisiers Saint-Aubin, CEDEX BP 48, 91192 Gif-sur-Yvette CEDEX, France; dominique.thiaudiere@synchrotron-soleil.fr (D.T.); cristian.mocuta@synchrotron-soleil.fr (C.M.)

**Keywords:** conjugated polymer and blends, in situ GIXD, additive, structure, strain

## Abstract

We describe the impact of tensile strains on the structural properties of thin films composed of PffBT4T-2OD π-conjugated polymer and PC_71_BM fullerenes coated on a stretchable substrate, based on a novel approach using in situ studies of flexible organic thin films. In situ grazing incidence X-ray diffraction (GIXD) measurements were carried out to probe the ordering of polymers and to measure the strain of the polymer chains under uniaxial tensile tests. A maximum 10% tensile stretching was applied (i.e., beyond the relaxation threshold). Interestingly we found different behaviors upon stretching the polymer: fullerene blends with the modified polymer; fullerene blends with the 1,8-Diiodooctane (DIO) additive. Overall, the strain in the system was almost twice as low in the presence of additive. The inclusion of additive was found to help in stabilizing the system and, in particular, the π–π packing of the donor polymer chains.

## 1. Introduction

One of the important research directions in the field of organic electronics is the development of stretchable/flexible electronics, which has attracted a lot of attention with a huge potential for numerous practical applications in different fields and daily life [1,2,3]. With the recent development of the flexible organic electronics, the performances of devices based on organic semiconducting molecules and polymers, such as organic light-emitting diodes (OLEDs), organic field effect transistors (OFETs) and organic solar cells (OSCs), have remarkably increased [4,5,6,7,8].

OSCs can be fabricated on large areas using low-cost roll-to-roll manufacturing techniques and, thus, represent an interesting alternative to conventional silicon solar cells [7,8], especially for nomadic applications and everyday connected objects, but also to construction, transport, urban furniture, etc. Recently, stretchable OSCs have demonstrated high potential as an energy source for the domain of flexible electronics [9,10,11]. The photoactive layer of an OSC is based on bulk heterojunctions, created by blending conjugated polymers and small molecule acceptors that provide low bandgap light absorption, high power conversion efficiency (PCE), appropriate energy level positions and quite high carrier mobilities [12,13]. In the last decade, tremendous efforts have been focused on the synthesis of new low bandgap donor polymers (e.g., PffBT4T-2OD polymer, commonly called PCE11) [14,15] and non-fullerene organic acceptors. The PCE of polymer-based solar cells has significantly increased and reached 16.5% in 2019 [16]. In most cases, additives, such as 1,8–diiodooctane (DIO), are used to improve nanoscale morphology and, thus, the performance of the organic solar cells [17,18]. DIO is a high boiling point additive used generally to dissolve PC_71_BM aggregates selectively [19], which improves the miscibility of fullerene molecules in PCE11. The additive slows down the formation of fullerene aggregates during the drying of the polymer blend, which makes the penetration of fullerene molecules between the chains of the donor polymer easier [20]. This effect was observed on the blend TQ1:PC_71_BM (TQ1:Poly[[2,3-bis(3-octyloxyphenyl)-5,8-quinoxalinediyl]-2,5-thiophenediyl]) [21] and on the PCE11:PC_71_BM mixture [19].

There are a lot of scientific efforts aimed at understanding and improving the microstructure of polymer-based OSCs, such as solution optimization [22,23], employing low volatile additives [24,25,26] and thermal annealing [27]. Moreover, the direct observation of film crystallization/formation during solvent evaporation gives insight into the phase separation in blends and allows correlating observed phenomenon to electrical properties [28], thus controlling polymer chain crystallization and fullerene molecule aggregation as a function of the blend concentration. In parallel, several methods have been developed with compact instrumentations, capable of tracking the crystallization process of semiconducting polymers and blends at synchrotron radiation facilities [8,29,30,31]

Taking into account this tremendous progress in understanding the structural evolution during the solution processing [31,32], the development of flexible and intrinsically stretchable organic conductors is very promising for the realization of emerging novel devices [33].

The commonly used approach is to probe thin films after stretching where the deformation of materials at a fixed value of strain can induce the anisotropy of the structural and morphological properties [34,35]. During the stretching process itself, the molecular structure and crystallinity can be strongly modified [36]. Moreover, if a phase transition takes place, it can dramatically influence the structural and electrical properties of the thin film [37]. Stretchable or wearable working devices rely on the knowledge and reproducibility of the evolution of electronic and optical properties under mechanical load [10,38,39]. It is worth noting that electrical transport measurements can also be a sensitive probe for strain-induced defects [40]. In our recent work, in situ grazing incidence X-ray diffraction (GIXD) studies were carried out to measure the structural parameters of pristine PCE11 polymer under uniaxial tensile load [41]. It was found that, after 15% of stretching a partial strain, relaxation takes place together with massive crack propagation. To go further towards relevant materials for OPV application, we expand on in situ GIXD metrology to the PCE11 blends with a focus on the impact of blending fullerenes inside the polymer on the mechanical properties. Furthermore, we put a particular focus on the role of additives under an applied strain in the pre-cracking regime, as additives not only improve device performances, but may also impact the nanoscale morphology and organization of the blended layers.

## 2. Materials and Methods

PCE11 (PffBT4T-2OD) (batch: YY11246CB) and PC_71_BM (99.5%) were purchased from 1-Material Inc. (Dorval, Quebec, Canada). PDMS stretchable substrate was prepared by spin coating of Sylgard 184 Silicone, obtained from Dow Corning (SAMARO, Lyon, France) using the ratio elastomer:hardener (10:1). After the degreasing of the polydimethylsiloxane (PDMS), using acetone, ethanol and water in an ultrasonic bath for 15 min for each bath and drying by argon flux gas, PDMS was fixed on a glass support, and the surface was activated by UV-ozone treatment at 80 °C for 10 min to improve the coatability.

The coating operation was carried out under an argon atmosphere inside a glove box. The blended layers with a thickness of 300 nm were spin coated at 1000 rpm from a chlorobenzene (CB):dichlorobenzene (DCB) (1:1) mixed solution of PCE11:PC_71_BM with a mixture ratio of 1:1.2 wt.% (33 mg/mL in total). For layers with 1,8-Diiodooctane (DIO), 3% DIO was added to the solution one hour before the coating.

The temperature of the polymer inks (with and without DIO) was kept at 110 °C and the substrate was heated at 100 °C during the spin coating operation. After coating, the layer was dried at 100 °C for 15 min and then the sample was immediately removed after drying.

Pure PCE11 layers were spin coated on glass and PDMS substrates at 1000 rpm from a CB solution of 15 mg/mL PCE11 concentration, the PCE11 ink was agitated at 110 °C for more than one hour in a glove box.

The structural properties of the thin flexible films were investigated by GIXD technique at two different beamlines:(i)DiffAbs beamline of SOLEIL synchrotron (Saint-Aubin, France) using a wide-area 2D XPAD detector (560 × 960 pixels of 130 μm) [42]. The measurements are recorded from the XPAD detector at different α_f_ ranges in the out-of-plane direction, as shown in Figure 1 (Positions 1 and 2). The distance sample-detector was 450 mm.(ii)BL9 beamline of DELTA synchrotron radiation facility at TU Dortmund, Dortmund, Germany, using a 2D image plate (MAR345) with a resolution of 100 µm/pixel [43]. The distance sample-detector was 394 mm.

For both experiments, X-ray photon energy of 15 keV was employed and a grazing incidence angle of 0.07° was chosen to probe the scattered signal from films. A photodiode point detector was used for aligning the sample at each stretching position.

A specially designed in situ stretching chamber allowed us to stretch thin films in a controlled way with a given amount of applied stretch and number of steps, as shown in Figure 1a. For each stretching step, the thin films were aligned prior to recording the GIXD patterns. The scattered signal was collected using a 2D detector. The sample surface was placed nearly horizontal, inclined by an incident angle α_i_ = 0.07°. The exit angle was denoted α_f_ at the azimuth angle φ. Figure 1a shows the out-of-plane (q_z_) and in-plane (q_xy_) directions of the two-dimensional GIXD pattern. The molecular structure of PCE11 and PC_71_BM molecules, and the schematic representation of the two main preferential orientations of the polymer chains (i.e., edge-on and face-on configurations with the backbone plane oriented perpendicular and parallel to the substrate, respectively), are depicted in Figure 1b.

## 3. Results and Discussions

### 3.1. Structural Properties before Stretching

Typically, spin coated semi-crystalline polymers favor edge-on orientation with the π–π stacking and conjugated backbone directions parallel to the substrate, whereas blending the donor polymer with fullerene acceptors leads to the co-existence of both edge-on and face-on orientations [19,20,44].

Our recent study on the structure of pristine PCE11 revealed that spin-coated layers on both rigid and stretchable substrates, are mostly edge-on oriented with more pronounced order and crystallinity for those coated on glass [41]. The present work is focused on PCE11 blends with fullerene molecules investigated via in situ GIXD under tensile stretching.

A 2D GIXD pattern for pristine PCE11 coated on glass is shown in Figure 2a. For pristine PCE11 film, the 2D pattern shows the edge-on orientation with the lamellar stacking perpendicular to the glass substrate. Such orientation results in a strong h00 series along the out-of-plane direction, and for in-plane direction, a pronounced 010 peak associated with π–π stacking, as shown in Figure 2a. The position of the 100 lamellar peak is at 3 nm^−1^ corresponding to a spacing of 2.09 nm, while the π–π stacking peak is centered at 17.5 nm^−1^ corresponding to a π–π spacing of 0.36 nm, which is in good agreement with the literature [15,45]. Initial GIXD patterns (before stretching) for the blends on PDMS are shown in Figure 2b,c. Interestingly, the lamellar stacking is flipped by 90°, resulting in dominating face-on orientation. In this case the lamellar 100 peak is mainly visible in-plane at 2.9 nm^−1^ (lamellar spacing of 2.17 nm) whereas 010 peaks appear only in the out-of-plane direction at 17.5 nm^−1^, as shown in Figure 2b,c. Additionally, broad PDMS and PC_71_BM halos, respectively, centered at 8.5 nm^−1^ and 13 nm^−1^ are visible.

Assuming the homogeneous distribution of lamellar spacings, the full width at half-maximum (FWHM) is mainly related to the finite domain size, as characterized by the coherence length, which is inversely proportional to the FWHM, according to the Scherrer equation [46]. Calculated domain sizes for each sample are given in Table 1. The FWHM corresponding to the in-plane 100 peak of pristine PCE11 is almost twice broader than for the PCE11:PC_71_BM blends. Similar behavior was observed comparing pure PCE11 polymer with PCE11:PCBM blends on hard supports [45]. In this study, the increase in the size of ordered domains was related to the presence of PCBM molecules in the PCE11 ordered phases, forming less perfect but larger area ordered domains.

Furthermore, our study is focused on the polymer:fullerene blends on the flexible substrates. To understand the role of the additive on the crystalline organization of the PCE11 and PC_71_BM blends, we have compared the line profiles extracted from 2D patterns, as shown in Figure 2b,c, along to the out-of-plane (q_z_) and in-plane (q_xy_) cuts. These profiles are shown in Figure 3a,b, respectively, and confirm dominating face-on orientation. The FWHM of the in-plane 100 peak for blend with DIO is ~1.2 larger than that for blend without DIO (same for the out-of-plane 010 peak), indicating smaller ordered domains, even though the degree of the π–π packing is improved after the addition of DIO, as shown in Table 1. This might also be due to the existence of bigger PC_71_BM molecules in the blend without DIO, forming less crystalline but larger area PCE11-ordered domains, as reported in [45]. With the addition of DIO, PC_71_BM aggregates dissolve better [19], leading to better crystallized but smaller PCE11 ordered domains.

A comparison of the azimuthal distribution for the blends processed with and without DIO is shown in Figure 3c,d, where the blend with DIO reveals more pronounced face-on orientation.

The ordering of the donor polymer is a very important parameter affecting the charge transport and interconnectivity within the blends, thus improving the solar cell performance [47]. The face-on crystallization of PCE11 chains is favorable for the charge transport in the direction across the active layer [15]. We observe here that face-on orientation as shown, for example, by the intensity of π−π stacking peak 010 is more pronounced for the blend film with DIO. Similar findings were reported in [48], supporting the idea that DIO is beneficial to the crystallinity of PCE11 chains in blends, resulting in better charge transport.

### 3.2. In Situ Structural Studies Under Stretching

In this section, we employ in situ GIXD metrology to monitor the microstructure of the polymer:fullerene blends upon stretching. Because of the increased scattering background on the flexible support, which makes it difficult to resolve the PCBM halo, we are focusing on the most intense PCE11 peaks. We compared the structural changes of the PCE11:PC_71_BM and PCE11:PC_71_BM with DIO blends upon stretching under grazing incidence geometry. The samples were uniaxially stretched during a tensile test starting from 1% with steps of 2% up to 7% and finally reaching 10%. For each step, the GIXD patterns have been recorded for the same angle of incidence of 0.07°. The maximum intensity of the in-plane 100 peak for the blended sample without additive was about two-times lower than for the one with DIO. To compensate the geometric effects and experimental constraints, we assume that the PDMS support provides an isotropic intensity distribution and the scattering ring from PDMS is independent of the azimuthal angle. Therefore, each pattern has been normalized on the corresponding PDMS ring. The normalized data show different trends for the blends processed with and without DIO upon stretching. Figure 4a shows the strain evolution (ε), calculated from Equation (1), of the π–π stacking distance d corresponding to the out-of-plane 010 peak position for the blends without and with DIO.
(1)$$\varepsilon \left[\%\right]=100\left(d-d_{0}\right)/d_{0}$$

Strain evolution appears clearly different for both blends—the PCE11:PC_71_BM processed without DIO shows an almost monotonous increase in tensile perpendicular strain up to 0.8% upon stretching. For the blend processed with DIO, the trend is opposite, resulting in a compressive perpendicular strain upon stretching.

Figure 4b shows the strain evolution of the polymer lamellar distance, corresponding to the in-plane 100 peak, for the blends with and without DIO. The strain evolution is almost similar for both blends. For the PCE11:PC_71_BM blend, there is an almost monotonous increase in tensile strain up to 0.8% upon stretching (the same order of magnitude found in a previous study for pristine PCE11 [41]), whereas for the blend with DIO, this increase is smaller by a factor of two. Comparing the features for the 010 and 100 peaks, their behaviors are different—the PCE11:PC_71_BM blend exhibits expansion in both directions. On the other hand, the PCE11:PC_71_BM blend with DIO shows a compressive out-of-plane strain. A possible stabilization of the systems during first few steps of the stretching might be a reason for the higher compressive strain at 3% of stretching.

The FWHM of the 010 peak for the blend with DIO remains unchanged upon stretching. For the blend without DIO we found a monotonous increase in the FWHM, reaching a similar value as that of the PCE11:PC_71_BM blend with DIO (not shown here). Similar to the 010 peak, the 100 FWHM for the PCE11:PC_71_BM blend without DIO remains unchanged upon stretching. For the blend with DIO we found a monotonous decrease in the FWHM, reaching a similar value as that of the PCE11:PC_71_BM blend.

From our recent findings on pristine PCE11 coated on PDMS [41], the initial orientation of PCE11 chains is mostly edge-on. In the present work, we found that blending PCE11 with PC_71_BM molecules leads to the reversed orientation (face-on). This face-on orientation becomes more pronounced after the addition of DIO. The switch in orientation from edge-on to face-on is clearly observed to influence the mechanical behavior of PCE11 chains. When the chains are edge-on oriented, tensile tests revealed that in the out-of-plane direction, the chains undergo increasing compressive strain (negative values of strains) until 10% of stretching, in agreement with the Poisson effect. On the other hand, the face-on oriented chains are undergoing expansion (positive values of strains) along both directions.

## 4. Conclusions

In comparison to pristine PCE11 films with dominating edge-on orientation, polymer:fullerene blends have shown a 90° switch with preferential face-on orientation. For the PCE11:PC_71_BM blends, the inclusion of the DIO additive further improves the microstructure and enhances face-on orientation.

In situ studies during tensile testing of PCE11:PC_71_BM processed with and without additives showed different mechanical behavior. In the case of the PCE11:PC_71_BM blend, only tensile strain has been observed, for the blend with DIO, we found a compressive out-of-plane strain associated with the π–π conjugation. It is also worth noting that the blends with DIO were almost by a factor of two less strained under the same stretching conditions. The implications of these findings, based on the inclusion of additives, could be beneficial for the flexibility and stability of organic photovoltaic devices. These in situ X-ray scattering studies during mechanical testing allow for the direct correlation of the structural and mechanical properties of organic materials for flexible electronics.

## Figures and Tables

**Figure 1 materials-13-03092-f001:**
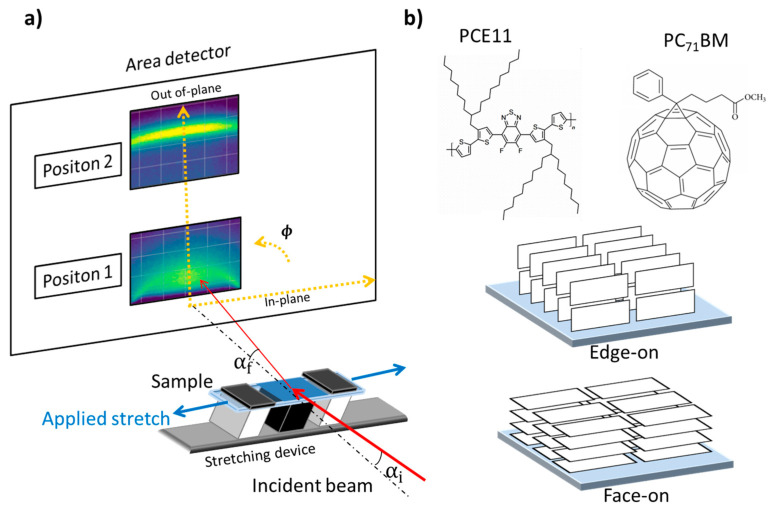
(**a**) Schematic view of the experimental setup for the GIXD experiments. (**b**) Molecular structure of PCE11 and PC_71_BM—schematic representation of edge-on and face-on orientations.

**Figure 2 materials-13-03092-f002:**
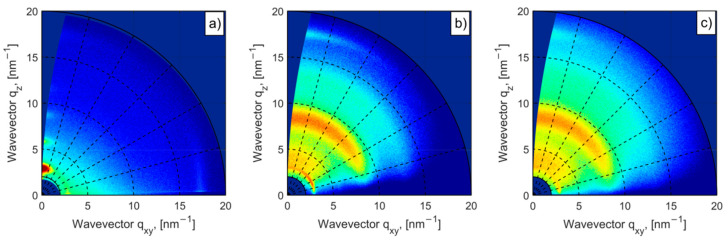
Two-dimensional GIXD patterns of (**a**) pristine PCE11 film coated on glass substrate and PCE11:PC_71_BM blend (**b**) with DIO, (**c**) without DIO, coated on PDMS substrates.

**Figure 3 materials-13-03092-f003:**
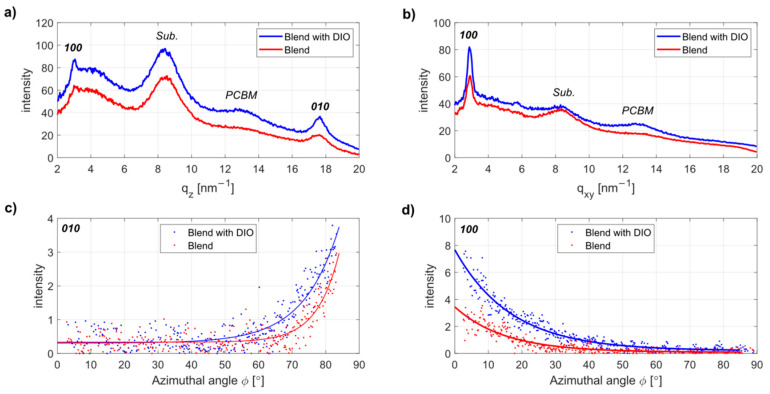
Out-of-plane (**a**) and in-plane (**b**) line profiles for PCE11:PC_71_BM blend (red circles) and PCE11:PC_71_BM blend with DIO (blue circles) on the PDMS; the azimuthal profiles for the 010 (**c**) and 100 (**d**) peaks, respectively.

**Figure 4 materials-13-03092-f004:**
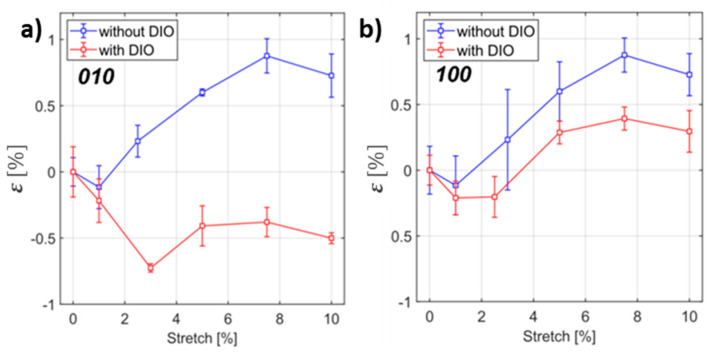
The strain evolution of the out-of-plane 010 π–π spacing (**a**) and in-plane 100 lamellar spacing (**b**) as a function of the applied stretch for the PCE11:PC_71_BM blends without (blue) and with DIO (red).

**Table 1 materials-13-03092-t001:** (100) and (010) domain sizes of pure PCE11 film, coated on glass, and PCE11:PC_71_BM films (with and without DIO additive) coated on PDMS.

Polymer	100 Domain Size (nm)		010 Domain Size (nm)	
	OOP	IP	OOP	IP
Pristine PCE11	12.9	9.6	4.46	4.22
Blend with DIO	11.3	11	6.75	-
Blend without DIO	13.56	13.2	8.1	-

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
