# Peer review of "Direct Observations of the Structural Properties of Semiconducting Polymer: Fullerene Blends under Tensile Stretching"

_materials, 2020, doi:10.3390/ma13143092_

Round 1

Reviewer 1 Report

The work presented by M.Y. Aliouat provides results about the effects of tensile strains on the structural properties of PffBT4T-2OD polymer and PC71BM fullerenes thin films using in-situ GIXD technique. Additionally they investigated the roles of 1,8-Diiodooctane (DIO) additive on strain behavior of the applied polymers. I believe the manuscript is easy to read and can be of interest for the potential readers and thus recommend its publication in Materials after a minor revision. There are only three comments give below that need to be addressed.

  1. The turning point, at which the trend for stretching changes has reported by the authors, J. Appl. Phys. 127, 045108 (2020), to be around 15%-20%. Why such a low (10%) tensile stretching is chosen for this study, while there would have been a significant part to investigate and thus share the outcomes with the potential readers. 
  2. Fig. 4: The blend with DIO shows a rapid decrease in tensile strain up to ~-0.7% after 3% stretch, while the strain increase to reach a plateau, following by a slight decline. The mechanism behind such trend needs to be explained.
  3. Fig. 1 and 4: Do not use right or left. please label the figures as alphabetically.

Author Response

1. In the previous study we found that that after a critical value of 10%–15% stretching, cracking occurs in the polymer layer. At the same time, the average elastic strain measured by GIXD decreases as a consequence of stress relaxation along the crack boundaries. The elastic strain energy stored in the PCE11 films decreases as a result of increasing crack length developed upon stretching higher than 15%.

Therefore, we have chosen for the current study an elastic regime in the range 0%–10% of stretching prior of the first cracking appearance. Higher strain and the cracking propagations are interesting issues and will be a subject of further investigations.

2. The system tries to stabilize during first few steps of the stretching and then getting to be aligned along stretching direction. This stabilization might be a reason of such increase of the strain. (updated in discussion part, lines 219-220)

3. We have modified Fig. 1 and 4 according to Reviewer suggestions

Reviewer 2 Report

The manuscript materials-852108 "Direct observations of the structural properties of semiconducting polymer: fullerene blends under tensile stretching" can be published in Materials after correction some minor text errors:

Line 56 To check if something is missing “    -5,8-dyl-alt-thiophene-2,5-diyl”

Line 76 Instead of [10][38,39], type [10, 38,39].

Line 133 The same for [19,20][44].

Line 148 [15][45] to [15, 45].

Line 241 Missed space after “:” in “…edge-on orientation polymer:fullerene…”

Author Response

We have made all changes requested by  Reviewer

Reviewer 3 Report

In this research, the authors describe the impact of tensile strains on the structural properties of thin films composed of PffBT4T-2OD π-conjugated polymer and PC71BM fullerenes coated on a stretchable substrate, based on a novel approach using in situ studies of flexible organic thin films. All materials have been carefully studied through two-dimensional GIXD patterns, and the line profiles extracted from 2D patterns. In addition, in situ experiment has been carried out during tensile testing of PCE11:PC71BM processed with and without additives. The research is carefully conducted, and all data are reasonable. I would like to recommend its publication after minor revisions.

  1. The ?[%] should also be provided in Figures 4 (right).
  2. Page 5, line 177,178 "A comparison of the azimuthal distribution for the blends processed with and without DIO is shown in Figure 3, c,d where the blend with DIO reveals more pronounced face-on orientation." the authors say it reveals more pronounced face-on orientation, why is the intensity in Figure 3, c so low?
  3. Please pay attention to your reference format, and the author should carefully check and revise the references.

[3]. Liquid-crystalline semiconducting polymers with high charge-carrier mobility. McCulloch, I., Heeney, M., Bailey, C. et al. 2006, Nature Mater 5, pp. 328–333. DOI: 10.1038/nmat1612.

[11].Flexible, highly efficient all-polymer solar cells. Taesu Kim1, Jae-Han Kim, Tae Eui Kang, Changyeon Lee, Hyunbum Kang, Minkwan Shin. 2015, Nat. Commun., p. 8547. DOI : 10.1038/ncomms9547.

Author Response

1. We provided   % sign in Figure 4.

2. In Figures 3,c  are shown the line profiles for π−π stacking 010 peak. This peak is rather at the high q-values (see Fig. 2, b,c), is relatively weak due to strong background of the scattering from the flexible support. However, approaching to the out-of-plane direction at 90° we can see an increase of intensity for the blend with DIO. Moreover, if we compare the line profiles for more intense 100 peaks, this trend is getting more pronounced, a twice increase of intensity for blend with DIO for in-plane direction at 0°.

3. We have checked all references and revised ref [3] and [11] in the manuscript.